# SeedLoRA: A Fusion Approach to Efficient LLM Fine-Tuning

**Yong Liu** [1] **Di Fu** [1] **Shenggan Cheng** [1] **Zirui Zhu** [1] **Yang Luo** [1] **Minhao Cheng** [2] **Cho-Jui Hsieh** [3] **Yang You** [1]

## Abstract

Despite Low-Rank Adaptation (LoRA)'s popularity for fine-tuning large models, it often exhibits a noticeable performance gap compared to full fine-tuning, particularly in complex tasks such as mathematical reasoning and code generation. We propose SeedLoRA, a novel fusion approach that bridges this gap by leveraging complementary strengths of multiple LoRA models trained with different random seeds on the same task. Unlike existing model merging methods that focus on combining knowledge from different tasks, SeedLoRA introduces a two-stage fusion strategy specifically designed for single-task scenarios: first identifying and preserving strong shared patterns across models, then performing principled subspace fusion in a unified representation space. Comprehensive experiments on LLaMA2-7B and Mistral-7B demonstrate that SeedLoRA significantly improves performance over individual LoRA models by 4.9% on GSM8K and 6.6% on HumanEval, effectively matching or exceeding full fine-tuning performance while maintaining the efficiency benefits of LoRA. Our analysis reveals that this improvement stems from Seed-LoRA's ability to effectively combine complementary strengths learned by different seeds in a common representation space.

## 1. Introduction

Parameter-Efficient Fine-Tuning (PEFT) methods have emerged as promising training schemes in fine-tuning large language models (LLMs), offering a balance between performance and efficiency. Among these, LoRA (Hu et al., 2022) has gained popularity due to its effectiveness and simplicity. Despite its advantages, LoRA often exhibits noticeable performance gap compared to full fine-tuning approaches, limiting its applicability in scenarios where state-of-the-art performance is required.

Researchers have proposed various approaches to narrow the performance gap between LoRA and full fine-tuning in LLMs. These methods typically fall into three categories: increasing LoRA's capacity, optimizing LoRA's structure, and combining multiple LoRA adaptations. For instance, ReLoRA (Lialin et al., 2024) proposes periodically increasing the rank during training, while DoRA (yang Liu et al., 2024) and MiLoRA (Wang et al., 2024a) suggest alternative low-rank structures and initialization strategies. Techniques such as MultiLoRA (Wang et al., 2023b) and MoLoRA (Zadouri et al., 2024) attempt to leverage multiple LoRA modules, inspired by Mixture of Experts models. While these approaches have shown improvements, they often come at the cost of increased computational complexity or fail to fully close the gap with full fine-tuning, particularly in challenging domains like mathematical reasoning and code generation.

In our investigation of these limitations, we made a key observation: models trained on identical tasks with different random seeds exhibit similar overall performance, yet demonstrate varying proficiency across different subdomains of the task. This opens the opportunity to combine their strengths into a more robust model.

Inspired by this insight, we naturally turn to model merging techniques, which have gained significant attention in the field of LLMs as means to combine knowledge from multiple models without increasing inference costs. However, we find that applying existing merging methods to our scenario presents unique challenges. Specifically, most existing work on model merging focuses on multi-task scenarios, aiming to integrate capabilities from models trained on different tasks (Wortsman et al., 2022; Ilharco et al., 2022). In contrast, our experiments reveal that the challenges faced in single-task model merging—our focus—differ substantially from those in multi-task scenarios. To elucidate this distinction, our analysis of cosine similarities reveals a crucial difference: models trained on different tasks exhibit near-zero similarity, indicating orthogonality, which leads

---

[1]Department of Computer Science, National University of Singapore [2]College of Information Sciences and Technology, Pennsylvania State University [3]Department of Computer Science, University of California, Los Angeles. Correspondence to: Yong Liu <liuyong@comp.nus.edu.sg>, Yang You <youy@comp.nus.edu.sg>.

*Proceedings of the 42nd International Conference on Machine Learning*, Vancouver, Canada. PMLR 267, 2025. Copyright 2025 by the author(s).

to interference issues in multi-task merging. Conversely, models trained on the same task with different seeds show high cosine similarity, suggesting a high degree of shared information. This fundamental difference shifts the primary challenge in single-task merging from interference mitigation to effective information combination and redundancy elimination, necessitating a new approach tailored specifically to single-task model merging.

Building on these insights, we propose SeedLoRA, a novel approach to address the unique challenges of single-task model merging. Our approach capitalizes on the high cosine similarity and shared information between models trained on the same task with different seeds, focusing on effective information combination and redundancy elimination.

At the core of SeedLoRA is a two-stage merging strategy that addresses the distinct aspects of model fusion. The first stage focuses on identifying and handling extreme parameter values - both consistently large magnitudes and conflicting directions - across different seed-specific models. This stage ensures robust integration of strongly learned patterns while preventing destructive interference. The second stage employs a sophisticated SVD-based fusion approach for the remaining parameters, operating in a shared subspace to effectively combine complementary information while eliminating redundancy. By decomposing the merging process into these distinct stages, our method can better preserve the unique strengths of individual models while creating a more robust combined representation.

Our experimental results demonstrate the effectiveness of SeedLoRA. By merging multiple LoRA models with a rank of 8, we achieve performance comparable to full fine-tuning in challenging tasks such as mathematical reasoning and code generation. This approach not only narrows the performance gap between LoRA and full fine-tuning but also preserves the efficiency advantages of PEFT methods.

The main contributions of this paper are:

- A comprehensive analysis of the performance characteristics of LoRA models trained with different seeds on the same task, revealing their complementary strengths in various subdomains.

- Insights into the fundamental differences between single-task and multi-task model merging, highlighting the need for specialized approaches in each scenario.

- The introduction of SeedLoRA, a novel intra-task model merging method that effectively combines information from multiple models while eliminating redundancy.

- Extensive empirical evidence demonstrating the effectiveness of SeedLoRA in narrowing the performance gap between LoRA and full fine-tuning, particularly in complex tasks like mathematical reasoning and code generation.

## 2. Preliminaries and Related Work

### 2.1. LoRA

LoRA enables efficient LLM fine-tuning by introducing trainable low-rank matrices while keeping original weights frozen. For a pre-trained weight matrix $W \in \mathbb{R}^{d \times k}$, LoRA introduces the update $W' = W + BA$, where $B \in \mathbb{R}^{d \times r}$ and $A \in \mathbb{R}^{r \times k}$ are low-rank matrices with rank $r \ll \min(d, k)$. Recent research has expanded upon the LoRA framework, exploring various enhancement. These include novel initialization method of $A$ and $B$ matrices (MiLoRA (Wang et al., 2024a), Pissa (Meng et al., 2024a), LoRA-GA (Wang et al., 2024b)), higher-rank approaches (MoRA (Jiang et al., 2024), PeriodicLoRA (Meng et al., 2024b), ReLoRA (Lialin et al., 2024)), innovative structural modification(DoRA (yang Liu et al., 2024)), advanced training (LoRA+ (Hayou et al., 2024)).

Drawing inspiration from the Mixture of Experts (MoE) paradigm, this approach dynamically combines multiple LoRAs , each potentially specialized for different tasks or domains. Examples include MultiLoRA (Wang et al., 2023b), MoLoRA (Zadouri et al., 2024), LoRAHub (Huang et al., 2023), and HydraLoRA (Tian et al., 2024). By leveraging both LoRA's parameter efficiency and the adaptive capacity of expert models, Mixture of LoRA aims to create more versatile models that can perform effectively across a board range of tasks than single-adaptation LoRA implementations.

### 2.2. Model Merge

Model merging aims to combine the knowledge encoded in multiple trained models into a single, enhanced model. Current research in model merging focuses on two main areas: Multi-task Merging and Same/Similar-task Merging. Multi-task merging combines models trained on different tasks into a single model capable of performing multiple tasks, leveraging task-specific knowledge, and maintaining efficiency. Same/Similar-task merging, though less explored, focuses on combining models trained on identical or closely related tasks to enhance robustness and generalization, with studies showing improved performance on shifted data distributions. Most work in this area has been conducted in computer vision, leaving significant opportunities for application in fields like natural language processing. Although a range of methods (Ilharco et al., 2022; Lu et al., 2024; Verma & Elbayad, 2024; Huang et al., 2024; Salamanca et al.; Tam et al.; Deep et al., 2024; Lu et al., 2024) for model merging have been proposed, this paper focuses on a selected set of methods that provide distinct ways of combining model parameters. Model soup (Wortsman et al., 2022) improves the accuracy of fine-tuned models by averaging the weights of multiple models fine-tuned with different hyperparameters, rather than selecting only the best individual model.

# 3. Proposed Method

## 3.1. Motivation

**Performance Gap between LoRA and Full Fine-Tuning.**

Prior research (Biderman et al., 2024) shows LoRA's performance gap persists even with increased rank or extended training. In Table 1, we conducted the experiments fine-tuning LLaMA2-7B with LoRA from rank=8 to rank=64 on math (MetaMathQA) and code generation (CodeFeedback) tasks (Yu et al., 2023). The results reveal that although increasing the rank value of LoRA from 8 to 64 improves performance on GSM8K and MATH from 39.64 to 41.05, a significant gap remains when compared with Full Fine-Tuning. A similar trend was observed in code generation tasks, where increasing the rank initially improves performance, but at extremely high ranks, such as 64, performance begins to decline. This indicates that while higher ranks can lead to gains, LoRA still struggles to fully match the performance of full fine-tuning, particularly in complex domains such as mathematical reasoning and code generation. These observations motivate the need for a novel training strategy and optimization method to further narrow the performance gap, enhancing LoRA's applicability across a wider range of tasks.

| Task | r=8 | r=16 | r=24 | r=32 | r=64 | FT |
|------|-----|------|------|------|------|-----|
| **GSM8K** | 64.0 | 65.6 | 64.9 | 64.7 | 65.6 | 66.5 |
| **MATH** | 15.3 | 15.3 | 16.3 | 16.6 | 16.5 | 19.8 |
| **Average** | 39.6 | 40.4 | 40.6 | 40.7 | 41.1 | 43.2 |

*Table 1.* Fine-Tuning LLaMA-2-7B model with LoRA on Meta-MathQA (seed=11).

## 3.2. Narrowing the Performance Gap via Model Merging

**Analyzing LoRA and Full Fine-Tuning Performance.** We conducted a comprehensive analysis to better understand the performance discrepancy between LoRA and full fine-tuning across various subdomains. Our approach involves visualizing the performance of multiple models trained with different random seeds using both LoRA and full fine-tuning techniques. Specifically, we leverage the Massive Multitask Language Understanding (MMLU) benchmark, which covers a wide range of subjects and allows for fine-grained performance analysis. Figure 1 illustrates the performance of LoRA and full fine-tuning models across different MMLU subdomains. The results reveal an interesting pattern: while LoRA models generally underperform compared to full fine-tuning, they exhibit competitive performance in specific subdomains. This nuanced performance distribution led to a

key observation: different LoRA models, each trained with unique random seeds, tend to excel in distinct subdomains. Building on this insight, we formulated a hypothesis: by strategically merging multiple LoRA models, each is trained with a distinct seed and with its own specialized strengths, we could potentially achieve performance comparable to full fine-tuning.

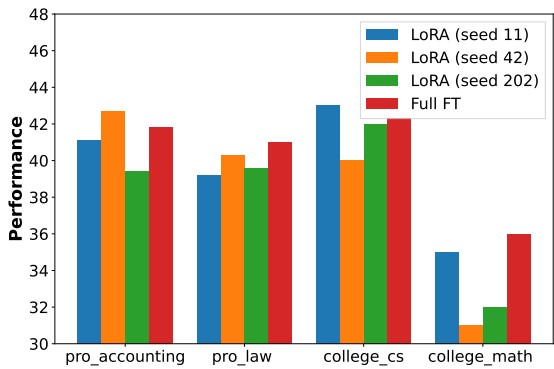

*Figure 1.* Performance comparison of LoRA and Full FT across MMLU subdomains.

**The Definition of Single-Task model Merging.** To explore our hypothesis of combining LoRA models with diverse strengths, we turn to the concept of model merging, which is the process of combining multiple models to enhance overall performance. While model merging is typically applied to integrate models trained on different tasks, we propose a novel application: merging models trained on a single task with different random seeds to achieve superior performance within that task. The proposed method is defined as follows:

Let $\theta_{pre}$ be the pre-trained base model, and $\{\theta_1, \theta_2, \ldots, \theta_n\}$ be a set of $n$ models fine-tuned on the same task using LoRA, each with a different random seed $s_i$. Each fine-tuned model $\theta_i$ can be represented as $\theta_i = \theta_{pre} + \tau_i$, where $\tau_i$ is the LoRA delta model for the $i$-th fine-tuned model, obtained using seed $s_i$. We then aim to merge these seed-specific delta models:

$$\tau_m = \text{Merge}(\tau_1, \tau_2, \ldots, \tau_n). \quad (1)$$

The merging is performed layer-wise for each LoRA adapter:

$$\tau_m^{(\ell)} = \text{Merge}(\tau_1^{(\ell)}, \tau_2^{(\ell)}, \ldots, \tau_n^{(\ell)}), \quad (2)$$

where $\tau_i^{(\ell)}$ represents the $\ell$-th layer of the $i$-th delta model trained with seed $s_i$. The final merged model is obtained by:

$$\theta_m = \theta_{pre} + \tau_m = \theta_{pre} + \text{Merge}(\tau_1, \tau_2, \ldots, \tau_n). \quad (3)$$

This approach leverages the diverse strengths of multiple LoRA models, each potentially excelling in different subdomains, to create a merged model that approaches or even

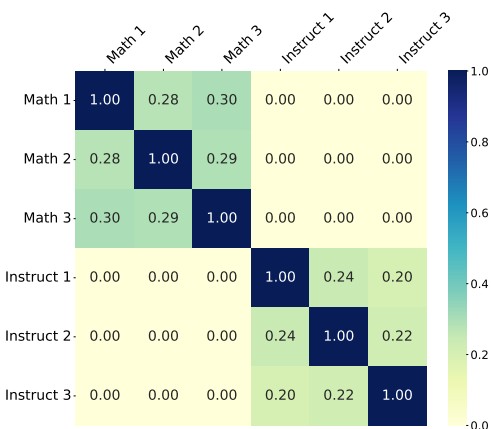

*Figure 2.* Cosine similarity comparison between LoRA delta models trained on the same and different tasks.

These fundamental differences explain why current mainstream multi-task merging techniques perform suboptimally in single-task scenarios. Methods like TIES-Merging and DARE primarily address interference from unrelated knowledge in orthogonal multi-task models. However, for highly similar and redundant single-task models, these strategies often fail to capture and integrate the subtle complementarity embedded within shared information. Their core design wasn't for extracting insights from high similarity, making them inefficient or ineffective here.

These limitations underscore the urgent need for a novel method tailored to single-task model merging. An effective strategy must leverage high knowledge sharing and manage **redundancy**, but also precisely extract and integrate **complementary** information from training randomness while mitigating minor **interference**.

### 3.4. SeedLoRA: A Two-Stage Approach for Model Merging

To tackle the redundancy, complementarity, and potential interference observed in single-task LoRA model merging, we propose SeedLoRA, a novel two-stage fusion strategy. This approach is designed to precisely integrate multiple LoRA adapters fine-tuned on the same task but initialized with different random seeds.

The first stage effectively manages inter-model redundancy and resolves potential interference by identifying and handling extreme parameter dimensions—those with consistently large magnitudes (robust) or opposing signs (conflicting). The second stage then targets the **residual dimensions**, which encompass all other parameters that survived the initial filtering. These residual dimensions contain subtle variations and complementary information that cannot be effectively merged through simple averaging, necessitating sophisticated subspace fusion techniques to extract and integrate their contributions within a unified low-rank representation space. Through this divide-and-conquer strategy, we can comprehensively leverage shared knowledge, uncover unique complementarities, and mitigate interference, thereby narrowing the performance gap between LoRA and full fine-tuning.

**Stage 1: Identifying Robust and Conflicting Dimensions.**

Given $n$ LoRA adapters, each adapter $\tau_i$ introduces low-rank parameter increments on top of a pre-trained model $\theta_{pre}$. Let $\tau_i(j)$ represent the value in dimension $j$ of the $i$-th adapter. We begin by defining a threshold $\sigma$ to mark "large" magnitudes, and we scan through every dimension $j$ across all $n$ adapters:

**Robust Dimensions.** If a sizeable subset of adapters exhibit $|\tau_i(j)| \geq \sigma$ and share the *same sign*, we mark these entries

surpasses the performance of full fine-tuning models. By focusing on models trained on a single task with different seeds, we capture a broader spectrum of task-specific knowledge while maintaining LoRA's efficiency advantages.

### 3.3. Difference between Multi-Task and Single-Task Model Merging

Before delving into our proposed single-task model merge method, we conducted a cosine similarity analysis to highlight its crucial differences from traditional multi-task approaches. To illustrate this, we trained 6 LoRA models: 3 for a mathematical reasoning task and 3 for an instruction following task, each with distinct random seeds (seed=1, 2, 3). Figure 2 shows our findings, highlighting a significant contrast in LoRA delta model ($\tau = BA$) cosine similarity between multi-task and single-task scenarios, profoundly impacting merging strategies:

- **Multi-task Scenarios:** Delta models from different tasks exhibit near-zero cosine similarity, indicating orthogonal learned knowledge. For multi-task merging, the main challenge is aggregating this diverse knowledge and resolving **interference** from unrelated information, aiming to retain unique contributions without compromising other tasks.

- **Single-task Scenarios:** In contrast, Delta models trained on the same task with different random seeds consistently show significantly positive cosine similarity. This reveals strong inherent correlation and high knowledge sharing (**redundancy**). Consequently, for single-task merging, the core challenge shifts to identifying and integrating subtle **complementary** information from training randomness. While **interference** risk exists, precisely extracting truly beneficial and unique contributions from this abundant, similar, and redundant information becomes critical.

as robust. Their consistent sign and large magnitude suggest that most adapters converge on a similar direction for that parameter. These robust dimensions represent the highly consistent shared core knowledge among models, illustrating their inherent redundancy. In practice, we average them to retain their collective strength, thereby effectively processing and leveraging this redundancy, and emphasizing the core shared patterns among models.

$$\tau_{\text{robust}}(j) = \frac{1}{|\mathcal{I}_j|} \sum_{i \in \mathcal{I}_j} \tau_i(j), \qquad (4)$$

where $\mathcal{I}_j$ is the set of adapters whose value in dimension $j$ exceeds $\tau$ and matches the dominant sign.

**Conflicting Dimensions.** If multiple distinct groups of adapters produce equally large but opposite-sign values in the same dimension, that dimension is regarded as a conflicting dimension. This indicates strong learning opposition in that specific parameter direction. If left unaddressed, this opposition will lead to negative interference at the model level, significantly undermining the effectiveness of straightforward averaging. Following related work on multi-task model merging (e.g., TIES (Yadav et al., 2023)), we resolve these conflicts by assigning them to one of the dominant sign groups (e.g., retaining only the majority sign). This step effectively prevents interference caused by parameter conflicts from contaminating subsequent processing, ensuring the robustness of the fusion.

**Stage 2: Subspace Fusion for Residual Dimensions**

After addressing parameter redundancy and interference in Stage 1, our second stage focuses on integrating complementary information from the remaining dimensions. These Residual Dimensions hold distinct, valuable contributions from different models. Since directly combining these unique patterns in their original high-dimensional space is challenging, our Subspace Fusion approach leverages SVD to align and combine their underlying complementary insights within a common subspace. This enables more precise identification and combination of complementary strengths, while also filtering out potential noise.

The specific process consists of the following steps:

**Step 1: Dimensionwise Averaging.**

For each layer $\ell$, consider the original updates $\tau_i^{(\ell)}$ from the $i$-th adapter. For each dimension $j$ that was not classified as robust or conflict in Stage 1 (i.e., a residual dimension), we build a single averaged value $M_{\text{avg}}^{(\ell)}$ by averaging across all adapters $i$:

$$M_{\text{avg}}^{(\ell)}(j) = \frac{1}{n} \sum_{i=1}^{n} \tau_i^{(\ell)}(j) \qquad (5)$$

where $n$ is the total number of adapters. This step collapses moderate differences into a single, consolidated update.

**Step 2: SVD-Based Low-Rank Decomposition.** While the averaged matrix $M_{\text{avg}}^{(\ell)}$ provides a consolidated representation, directly merging individual adapters in the original parameter space may not effectively capture their complementary strengths. SVD decomposition identifies the principal directions of variation across models and establishes a common coordinate system where model differences can be systematically compared and merged, enabling more precise fusion of complementary information.

We perform a truncated singular value decomposition on $M_{\text{avg}}^{(\ell)}$:

$$M_{\text{avg}}^{(\ell)} \approx U^{(\ell)} \Sigma^{(\ell)} V^{(\ell)\top}. \qquad (6)$$

The rank $r$ is chosen to capture the principal directions while discarding less important components. The matrices $U^{(\ell)} \in \mathbb{R}^{d \times r}$ and $V^{(\ell)} \in \mathbb{R}^{k \times r}$ (for a layer of shape $d \times k$) define a shared basis in the row and column spaces, respectively, whereas $\Sigma^{(\ell)}$ indicates the importance of each direction.

**Step 3: Re-Projection of Individual Adapters.** Each adapter's parameters $\tau_i^{(\ell)}$ can then be projected onto this common subspace. In this way, multiple single models can be aligned and easily edited or merged. Specifically, we compute a coordinate matrix:

$$Z_i^{(\ell)} = \left(U^{(\ell)}\right)^\top \tau_i^{(\ell)} V^{(\ell)}, \qquad (7)$$

which projects adapter $i$ to the common subspace. By this way, we can further edit or fuse these projected matrices $\{Z_i^{(\ell)}\}$ to merge the knowledge from different models in the next step.

**Step 4: Fusion and Reconstruction.** To form a single merged model for these dimensions in stage 2, we merge $\{Z_i^{(\ell)}\}$ into a fused set of coordinates $\widetilde{Z}^{(\ell)}$. Specifically, we reuse existing fusion methods like TIES, DARE, and Weighted Averaging. Finally, we reconstruct:

$$\tau_{\text{fused}}^{(\ell)} = U^{(\ell)} \widetilde{Z}^{(\ell)} V^{(\ell)\top}. \qquad (8)$$

This yields a single low-rank representation of the parameters in stage 2.

**Final Assembly of the Fused Adapter.** We assemble the final fused LoRA adapter $\tau_{\text{final}}$ by combining these subspace-fused residual parameters with the robust and conflict outcomes from Stage 1. In particular, for each layer $\ell$:

$$\tau_{\text{final}}^{(\ell)} = \tau_{\text{robust}}^{(\ell)} + \tau_{\text{conflict}}^{(\ell)} + \tau_{\text{fused}}^{(\ell)}, \qquad (9)$$

where $\tau_{\text{robust}}^{(\ell)}$ includes dimensions identified as consistently large and same-sign, $\tau_{\text{fused}}^{(\ell)}$ is the outcome of the SVD-

| | Method | seed 11 | seed 42 | seed 202 | SeedLoRA | Model Soup | TIES | DARE |
|---|---|---|---|---|---|---|---|---|
| | Evaluating LLaMA2-7B on GSM8K. The performance of Full Fine-Tuning is 66.5. | | | | | | | |
| **LLaMA2-7B** | **LoRA (r=8)** | 64.0 | 63.8 | 64.1 | **68.9** | 66.6 | 65.7 | 65.7 |
| | **LoRA+ (r=8)** | 64.4 | 64.7 | 65.4 | **69.8** | 67.0 | 63.2 | 56.7 |
| | **DoRA (r=8)** | 64.6 | 64.7 | 64.7 | **68.9** | 66.3 | 66.0 | 67.3 |
| | Evaluating LLaMA2-7B on MATH. The performance of Full Fine-Tuning is 19.8. | | | | | | | |
| **LLaMA2-7B** | **LoRA (r=8)** | 15.3 | 15.3 | 14.9 | **17.8** | 15.7 | 16.0 | 15.5 |
| | **LoRA+ (r=8)** | 15.4 | 15.5 | 16.0 | **18.2** | 16.1 | 16.7 | 16.4 |
| | **DoRA (r=8)** | 15.4 | 15.4 | 14.9 | **17.8** | 16.0 | 15.8 | 15.6 |
| | Evaluating Mistral-7B on GSM8K. The performance of Full Fine-Tuning is 78.6. | | | | | | | |
| **Mistral-7B** | **LoRA (r=8)** | 75.4 | 75.7 | 76.3 | **80.7** | 79.1 | 75.1 | 75.1 |
| | **LoRA+ (r=8)** | 76.5 | 73.5 | 75.9 | **80.3** | 79.7 | 79.4 | 78.7 |
| | **DoRA (r=8)** | 77.0 | 75.7 | 76.5 | **81.0** | 77.0 | 79.1 | 78.5 |
| | Evaluating Mistral-7B on MATH. The performance of Full Fine-Tuning is 28.5. | | | | | | | |
| **Mistral-7B** | **LoRA (r=8)** | 25.9 | 24.8 | 25.4 | **28.8** | 28.5 | 24.8 | 25.0 |
| | **LoRA+ (r=8)** | 25.1 | 25.2 | 25.4 | **28.0** | 27.9 | 25.9 | 24.3 |
| | **DoRA (r=8)** | 25.9 | 25.3 | 25.8 | **29.0** | 28.3 | 26.5 | 25.7 |

*Table 2.* Fine-Tuning LLaMA-2-7B and Mistral-7B with LoRA on MetaMathQA.

based merging on residual entries, and $\tau_{\text{conflict}}^{(\ell)}$ reflects any decisions made about sign-divergent dimensions.

Once aggregated across all layers, the result is a single low-rank update $\tau_{\text{final}}$ that retains the efficiency of LoRA while exploiting each adapter's advantages. Robust dimensions are preserved in their collectively strong direction, conflicts are neutralized to avoid destructive interference, and residual parameters are reconciled through a shared subspace.

## 4. Experimental Results

### 4.1. Experimental Setting

**Training and Evaluation:** For code generation, we use Code-Feedback (Zheng et al., 2024) as training data, LLaMA2-7B (Touvron et al., 2023) and Mistral-7B-v0.1 (Jiang et al., 2023) serve as base models. We evaluate using HumanEval (Chen et al., 2021), an established benchmark for Python text-to-code generation. For comprehensive assessment, we incorporate HumanEval+ from EvalPlus (Liu et al., 2024). For math reasoning, the Meta-MathQA (Yu et al., 2023) dataset is employed to fine-tune on the LLaMA2-7B and Mistral-7B models. The evaluation is conducted using the GSM8k (Cobbe et al., 2021) and MATH (Hendrycks et al., 2021) benchmarks, which are specifically constructed to test the model's capacity for mathematical reasoning and problem-solving. For the general domain, the TÜLU V2 (Wang et al., 2023a) dataset is

utilized in training on the LLaMA2-7B and Mistral-7B-v0.1. Following the setting of Open-Instruct (Ivison et al., 2023), we evaluate model on MMLU (Hendrycks et al., 2020), GSM8k, BBH (Suzgun et al., 2022), TyDiQA (Clark et al., 2020), TruthfulQA (Lin et al., 2021) and HumanEval.

### 4.2. Mathematical Reasoning

To validate the efficacy of our proposed merge method, we first evaluate the LoRA models with 3 different seeds on GSM8K and MATH, followed by an assessment of our merged model. The experimental results, shown in Table 2, demonstrate that the merged model substantially improve the performance of each independent model. Notably, for LoRA fine-tuning on LLaMA2-7B, SeedLoRA can improve the performance of vanilla LoRA from 64.1 to 68.9 on GSM8K and from 15.3 to 17.8 on MATH. Furthermore, to evaluate the generalizability, we extend our evaluation to LoRA variants (such as LoRA+ and DoRA) and more advanced pre-trained LLM (such as Mistral-7B). These additional experiments consistently demonstrate performance improvement when using our model merging approach. Finally, we also conduct experiments to compare SeedLoRA with current popular model merge methods, such as Model Soup (Wortsman et al., 2022), TIES (Yadav et al., 2023) and DARE (Yu et al., 2024). The experimental results on Table 2 illustrate these methods can also improve the performance of vanilla LoRA, but SeedLoRA can obtain more performance gain .

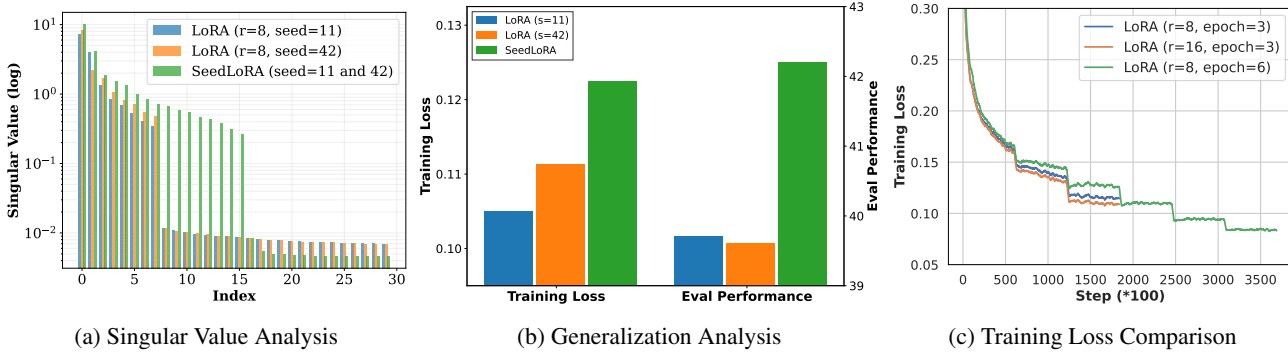

(a) Singular Value Analysis  (b) Generalization Analysis  (c) Training Loss Comparison

*Figure 3.* (a) Singular Value Analysis. (b) Generalization Analysis (c) Training Loss Analysis.

## 4.3. Code Generation

Building upon our findings in mathematical reasoning, we further evaluate the performance gain of our merged model on the Code Generation task. Table 3 presents the experimental results of individual training models and merged models on CodeFeedback benchmark. The data demonstrates that our merged model consistently outperforms individual models in the HumanEval and HumanEval+ tasks. Particularly, SeedLoRA exhibits exceptional performance on HumanEval (+) benchmark, surpassing the best individual LoRA by both 6.1% on LLaMA2-7B and Mistral-7B.

To contextualize our method's performance within a broader range of model merging approaches, we conducted a comparative analysis with popular approaches such as Model Soup, TIES and DARE. Our finding indicates our method achieves superior performance compared to these existing merge methods. For instance, our method enhances the performance of model soup from 34.1% to 40.2% for LoRA.

| | LoRA | SeedLoRA | MS | TIES | DARE |
|---|---|---|---|---|---|
| Full FT LLaMA2-7B on Humaneval: 40.3 | | | | | |
| **LoRA** | 34.1 | **40.7** | 34.1 | 38.4 | 36.0 |
| **LoRA+** | 36.6 | **39.2** | 39.0 | 32.3 | 30.5 |
| **DoRA** | 34.1 | **37.3** | 32.3 | 33.5 | 34.8 |
| Full FT LLaMA2-7B on Humaneval+: 37.1 | | | | | |
| **LoRA** | 30.5 | **36.6** | 29.9 | 34.1 | 30.5 |
| **LoRA+** | 34.1 | **36.6** | 34.1 | 28.7 | 27.4 |
| **DoRA** | **32.3** | **32.3** | 29.3 | 29.9 | 31.7 |

*Table 3.* LLaMA2-7B model with LoRA (Delta) on CodeFeedback (Humaneval and Humaneval+). MS represents Model Soup.

## 4.4. Instruction Following

Having examined the effectiveness of our proposed method SeedLoRA in specialized domains, we now extend our eval-

uation to general domain instruction tuning tasks. The experimental results, shown in the Table 4, demonstrate that our proposed method continues to improve upon the performance of the best individual model. However, the magnitude of improvement in this domain is less pronounced than observed in math reasoning and code generation. We believe this discrepancy arises from the nature of general domain tasks, where models are required to follow instructions rather than acquire new knowledge, as is often necessary for mathematical and coding tasks. Moreover, this observation underscores the efficacy of our method while also highlighting the challenges of achieving substantial gains in areas where LoRA already performs close to full fine-tuning.

## 4.5. Further Discussions

**Why Merging Models from the Same Task Can Improve the Performance?** To understand the performance improvements achieved by merging models from the same task (but different seeds), we conduct two key analyses: knowledge fusion and generalization ability.

Firstly, we evaluate whether the merged model can effectively fuse the knowledge from two individual models. We employ Singular Value Decomposition (SVD) to analyze the knowledge representation in each model. Figure 3a illustrates the singular value distribution of individual LoRA models (each with rank 8) and the merged SeedLoRA model. Notably, SeedLoRA exhibits a broader range of non-zero singular values compared to the individual LoRA, suggesting successful knowledge fusion from multiple sources.

Inspired by SWA (Izmailov et al., 2018), which claims that averaging weights can lead to wider optima and better generalization, we investigate whether our model exhibits similar benefits. We analyze the training loss and the performance on downstream evaluation tasks, as shown in Figure 3b and Figure 3c. Interestingly, SeedLORA demonstrates a slightly higher training loss but achieves superior evaluation performance on downstream evaluation tasks which takes different distributions from the training data. This

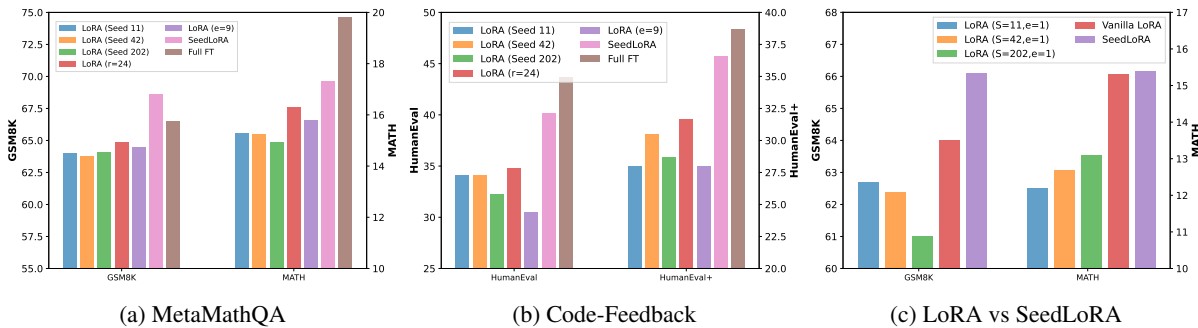

(a) MetaMathQA          (b) Code-Feedback          (c) LoRA vs SeedLoRA

*Figure 4.* The comparison between vanilla LoRA training with different seeds, with higher rank, with more epochs and Full Fine-Tuning (Full FT). (a) Comparison on MetaMathQA benchmark. (b) Comparison on Code-Feedback benchmark. (c) The Performance Comparison between LoRA and Seed LoRA under the similar training cost constraint.

| | Method | MMLU | GSM8K | BBH | TyDiQA | HumanEval | Average |
|---|---|---|---|---|---|---|---|
| **LoRA** | **LoRA (r=8)** | 49.2 | 22.5 | 43.3 | 51.8 | 14.9 | 36.4 |
| | **SeedLoRA** | 49.9 | 22.0 | 46.3 | 52.7 | 15.5 | **37.3** |
| | **Model Soup** | 50.3 | 20.5 | 45.5 | 50.9 | 15.2 | 36.5 |
| | **TIES** | 49.3 | 19.5 | 43.3 | 53.6 | 15.1 | 36.2 |
| | **DARE** | 50.2 | 22.5 | 44.3 | 53.4 | 15.3 | 37.0 |
| **LoRA+** | **LoRA+ (r=8)** | 49.7 | 25.0 | 46.5 | 53.1 | 16.0 | 38.1 |
| | **SeedLoRA** | 51.0 | 25.5 | 46.9 | 52.8 | 18.0 | **38.8** |
| | **Model Soup** | 51.2 | 24.5 | 45.7 | 52.7 | 17.4 | 38.3 |
| | **TIES** | 50.2 | 23.0 | 42.9 | 52.8 | 17.5 | 37.3 |
| | **DARE** | 50.1 | 22.0 | 43.1 | 52.9 | 17.4 | 37.1 |
| **DoRA** | **DoRA (r=8)** | 49.4 | 25.5 | 46.3 | 50.4 | 16.0 | 37.5 |
| | **SeedLoRA** | 50.0 | 28.5 | 46.6 | 52.2 | 15.2 | **38.5** |
| | **Model Soup** | 50.4 | 23.0 | 47.7 | 51.0 | 15.2 | 37.5 |
| | **TIES** | 49.8 | 22.5 | 45.6 | 53.1 | 14.9 | 37.2 |
| | **DARE** | 49.7 | 23.5 | 45.3 | 53.3 | 14.7 | 37.3 |

*Table 4.* LLaMA-2-7B model with LoRA on Tulu-v2. For the results of LoRA and its variants, we report the best performance of 3 LoRA models, which is trained with different seeds.

pattern indicates improved generalization ability, suggesting SeedLoRA learns more robust, task agnostic features rather than overfitting the training data. These findings on knowledge fusion and generalization provide insight into the mechanisms underlying SeedLoRA's improved performance across various tasks.

**Comparing with Higher Rank LoRA.** To further validate the effectiveness of our approach, we compare SeedLoRA with higher rank LoRA models. The main reason is that we are training 3 models, so we compare with training one LoRA with rank 8*3. Since our experiments focus on merging three LoRA models with rank 8, we perform an ablation study comparing our merged model with a single LoRA model with rank 24. As shown in Figure 4, SeedLoRA outperforms the higher-rank LoRA model. This highlights

the advantage of SeedLoRA in effectively combining the strengths of multiple lower-rank models, achieving better performance than simply increasing the rank of a single model.

**Training SeedLoRA with Similar Cost as Formal LoRA.** Our method requires obtaining several LoRA models trained on the same tasks. While some suitable models can be found on platforms like Huggingface, it is often necessary to train multiple models by ourselves, potentially incurring additional training time. To address this, we investigate whether we can achieve better performance with comparable training cost using SeedLoRA.

Specifically, we propose an alternative to the standard 3-epoch LoRA fine-tuning of LLaMA2-7B: training 3 individual models with 1 epoch each, then merging these 3

partially trained models. We conduct this experiment on MetaMathQA benchmark, with the results shown in Figure 4(c). Remarkably, this approach outperforms the standard 3-epoch training while maintaining the same overall training time. This finding suggests a potentially new, more efficient training paradigm for PEFT.

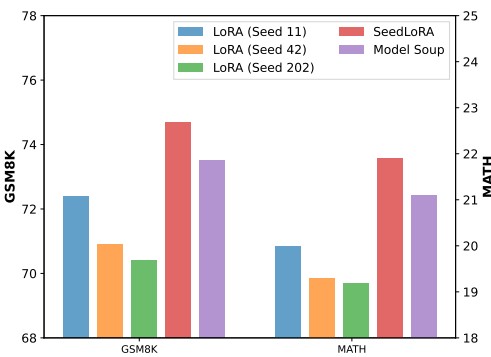

*Figure 5.* Scaling Results of LLaMA2-13B on MetaMathQA.

**Training with More Epochs.** The Training paradigm of SeedLoRA can be regarded as training LoRA with more epochs. To rigorously validate the superior performance of SeedLoRA, we train vanilla LoRA with more epochs and compare our merged model with it. We conduct the experiment on MetaMathQA and Code-Feedback and the comparison result is shown in the Figure 4 and Figure 3c. The results illustrate that SeedLoRA can outperform LoRA training with more epochs on both math reasoning and code generation tasks, although training more epochs can slightly improve its performance.

**Scaling Results.** To verify the scalability of SeedLoRA, we conduct experiments on pre-trained models with larger number of parameters. Specially, we evaluate the performance of SeedLoRA on the LLaMA2-13B model, The results are presented in Figure 5. SeedLoRA achieves approximately 2.3% performance gain compared to the best individual LoRA model. This demonstrates that SeedLoRA can effectively improve the performance even on larger models, highlighting its scalability and potential for enhancing models across different sizes.

## 5. Conclusions

In this paper, we introduce SeedLoRA, a novel single-task model merging approach designed to enhance LoRA fine-tuning. Our method effectively narrows the performance gap between LoRA and full fine-tuning in complex tasks like mathematical reasoning and code generation by combining complementary strengths of models trained with different seeds. Notably, SeedLoRA consistently outperforms existing merging techniques in single-task scenarios. The effectiveness of SeedLoRA stems from its ability to

fuse knowledge from individual models that specialize in different sub-domains, leading to improved generalization. By bridging the performance gap between PEFT methods and full fine-tune, our work highlights the potential to enable broader adoption of state-of-the-art LLMs in resource-constrained environments.

## Impact Statement

This paper presents work whose goal is to advance the field of Machine Learning. There are many potential societal consequences of our work, none which we feel must be specifically highlighted here.

## Acknowledgements

Yang You's research group is being sponsored by NUS startup grant (Presidential Young Professorship), Singapore MOE Tier-1 grant, ByteDance grant, ARCTIC grant, SMI grant and Alibaba grant. This work is also supported by NSF 2048280, 2325121, 2244760, 2331966 and ONR N00014-23-1- 2300:P00001.

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

# A. Appendix

## A.1. Implementation Details

Training is conducted on Nvidia A100 and H100 GPUs using BFloat16 precision. We set weight decay to 0 and employ a cosine learning rate scheduler with a 0.03 ratio linear warmup. For evaluation, we utilize vLLM (Kwon et al., 2023) to conduct our tests, ensuring efficient and scalable inference.

| Model | Dataset | Method | r | $\alpha$ | LR | LR Scheduler | Warmup | Epochs | Batch Size | $\sigma$ |
|---|---|---|---|---|---|---|---|---|---|---|
| **LLaMA2-7B** | MetaMathQA | LoRA | 8 | 16 | 3e-5 | cosine | 300 | 3 | 128 | median |
| **LLaMA2-7B** | Tulu-v2 | LoRA | 8 | 16 | 3e-5 | cosine | 500 | 2 | 128 | median |
| **LLaMA2-7B** | Code-Feedback | LoRA | 8 | 16 | 3e-5 | cosine | 300 | 3 | 32 | median |
| **Mistral-7B** | MetaMathQA | LoRA | 8 | 16 | 3e-5 | cosine | 300 | 3 | 128 | median |
| **Mistral-7B** | Tulu-v2 | LoRA | 8 | 16 | 3e-5 | cosine | 500 | 2 | 128 | median |
| **Mistral-7B** | Code-Feedback | LoRA | 8 | 16 | 5e-5 | cosine | 300 | 3 | 32 | median |

*Table 5.* LLaMA-2-7B model with LoRA on Tulu-v2. For the results of LoRA and its variants, we report the best performance of 3 LoRA models, which is trained with different seeds.

## A.2. The Performance Analysis about LoRA and Full Fine-Tuning on MBPP

In Table 6, we conducted the experiments fine-tuning LLaMA2-7B with LoRA from rank=8 to rank=64 on code generation (CodeFeedback) tasks (Yu et al., 2023). The results illustrate that although increasing the rank value of LoRA from 8 to 64 improves performance on HumanEval from 34.1 to 35.4, a significant gap remains when compared with Full Fine-Tuning.

| Task | rank=8 | rank=16 | rank=24 | rank=32 | rank=64 | Full FT |
|---|---|---|---|---|---|---|
| **HumanEval** | 34.1 | 34.1 | 34.8 | 34.8 | 35.4 | 40.3 |
| **HumanEval+** | 28.0 | 32.3 | 31.7 | 31.7 | 31.7 | 37.1 |
| **MBPP (+)** | 45.8 (38.6) | 43.7 (36.0) | 44.2 (36.2) | 46.6 (39.7) | 42.1 (36.2) | 53.1 |
| **Average** | 40.0 (33.3) | 38.9 (34.2) | 39.5 (34.0) | 40.7 (35.7) | 38.8 (34.0) | 46.7 |

*Table 6.* LLaMA-2-7B model with LoRA (Delta) on Code-Feedback (seed=11).

## A.3. The experimental Results for Fine-Tuning Mistral-7B on Code-Feedback.

Table 7 presents the experimental results of Mistral-7B model with LoRA on CodeFeedback. The results demonstrate that our merged model consistently outperforms individual models in the HumanEval and HumanEval+ tasks.

| | Method | seed 11 | seed 42 | seed 202 | SeedLoRA | Model Soup | TIES | DARE |
|---|---|---|---|---|---|---|---|---|
| | **LoRA (r=8)** | 53.0 | 51.8 | 48.2 | **58.0** | 53.7 | 55.5 | 56.7 |
| **HumanEval** | **LoRA+ (r=8)** | 54.3 | 48.8 | 47.6 | **56.9** | 54.3 | 51.8 | 54.3 |
| | **DoRA (r=8)** | 54.3 | 55.5 | 45.1 | **57.6** | 54.3 | 56.7 | 55.5 |
| | **LoRA (r=8)** | 49.4 | 47.6 | 40.9 | **51.2** | 50.6 | 49.4 | 50.0 |
| **HumanEval+** | **LoRA+ (r=8)** | 48.2 | 43.9 | 40.2 | **49.4** | 48.2 | 47.6 | 48.8 |
| | **DoRA (r=8)** | 47.6 | 49.4 | 42.1 | **49.6** | 48.8 | 51.8 | 50.2 |

*Table 7.* Mistral-7B model with LoRA (Delta) on CodeFeedback (HumanEval and HumanEval+).

### A.4. The experimental Results for Fine-Tuning Mistral-7B on Tulu-v2

Table 8 demonstrates that our proposed method on Tulu-v2 can continue to improve upon the performance of the best individual model with fine-tuning Mistral-7B.

|  | MMLU-0 | GSM8K | BBH | TyDiQA | HumanEval | Average |
|---|---|---|---|---|---|---|
| **LoRA (r=8)** | 59.4 | 46.0 | 55.0 | 59.9 | 33.8 | 50.8 |
|  | 58.8 | 44.0 | 56.6 | 59.5 | 35.5 | 50.9 |
|  | 58.2 | 50.5 | 58.7 | 59.0 | 31.4 | 51.6 |
| **SeedLoRA** | 61.0 | 52.0 | 59.6 | 62.0 | 33.2 | **53.6** |
| **TIES** | 58.3 | 42.5 | 53.8 | 60.4 | 33.7 | 49.7 |
| **DARE** | 58.5 | 42.0 | 56.2 | 60.5 | 35.4 | 50.5 |
| **LoRA+ (r=8)** | 60.8 | 45.0 | 59.4 | 58.2 | 34.1 | 51.5 |
|  | 61.2 | 45.5 | 59.7 | 59.7 | 32.2 | 51.7 |
|  | 60.5 | 47.0 | 58.6 | 59.1 | 32.0 | 51.4 |
| **SeedLoRA** | 61.6 | 47.0 | 60.8 | 60.2 | 35.2 | **53.0** |
| **TIES** | 60.6 | 46.0 | 57.9 | 58.8 | 34.1 | 51.5 |
| **DARE** | 60.4 | 41.0 | 56.3 | 59.2 | 34.2 | 50.2 |
| **DoRA (r=8)** | 61.1 | 46.0 | 58.8 | 58.9 | 34.3 | 51.8 |
|  | 60.3 | 52.0 | 58.8 | 60.1 | 33.1 | 52.9 |
|  | 60.3 | 52.0 | 58.5 | 59.9 | 32.9 | 52.7 |
| **SeedLoRA** | 62.0 | 51.5 | 61.0 | 61.2 | 34.2 | **54.0** |
| **TIES** | 60.5 | 46.5 | 58.2 | 58.2 | 35.3 | 51.8 |
| **DARE** | 60.5 | 44.0 | 57.4 | 59.3 | 35.6 | 51.4 |
| **LoRA (r=24)** | 60.4 | 46.5 | 57.4 | 59.6 | 31.8 | 51.1 |
| **LoRA (epoch=6)** | 56.5 | 47.0 | 54.0 | 55.5 | 32.5 | 49.1 |

*Table 8.* Mistral-7B model with LoRA (Delta) on Tulu-v2.

### A.5. The Number of Models

To investigate scaling behavior when merging additional models, we conducted experiments merging up to 6 different seed models in Table 9. We should also mention that these results might vary slightly from previous findings due to differences in experimental environments. These results show significant initial gains when merging 2 models (+3.7% on GSM8K), with continued improvements up to 4 models (+5.7% on GSM8K). Beyond 4 models, we observe diminishing returns without performance degradation.

| Task | N=1 | N=2 | N=3 | N=4 | N=5 | N=6 |
|---|---|---|---|---|---|---|
| **GSM8K** | 64.0 | 67.7 | 68.4 | 69.7 | 69.6 | 69.8 |
| **MATH** | 15.3 | 16.6 | 17.3 | 17.1 | 16.6 | 17.1 |
| **Average** | 39.7 | 42.2 | 43.0 | 43.4 | 43.1 | 43.5 |

*Table 9.* LLaMA2-7B on MetaMathQA with More Seed Models.

