# OpenReview forum: "SeedLoRA: A Fusion Approach to Efficient LLM Fine-Tuning"
_ICML.cc/2025/Conference — ICML 2025 poster_

### Official Review · Reviewer_7HoA · 2025-02-25

**Overall Recommendation:** 3

**Summary:**

The paper presents a method that leverages multiple LoRA models trained with different random seeds on the same task and merges their trained weights using a two-stage merging strategy. In the first stage, the algorithm detects robust and conflicting dimensions from the multiple trained weights using thresholding and counting opposite-sign values, respectively. In the second stage, the algorithm calculates an average space using SVD and projects the weight matrices onto this space. The projected coordinates are then fused and used to reconstruct the fused weight matrix. The final adapter is formed by combining the robust dimensions, conflicting dimensions, and the reconstructed fused matrix. The experimental results demonstrate significant improvements in mathematical reasoning and code generation tasks over individual LoRA models, often matching or exceeding full fine-tuning performance.

**Claims And Evidence:**

The key claims made in the paper are well-supported by empirical evidence:
- The claim that SeedLoRA improves performance over individual LoRA models is substantiated through rigorous benchmarking on LLaMA2-7B and Mistral-7B models, showing improvements of up to 4.9% on GSM8K and 6.6% on HumanEval.
- The paper convincingly argues that models trained with different seeds exhibit complementary strengths, and merging them can lead to a more robust final model.

**Essential References Not Discussed:**

The paper provides a well-rounded discussion of prior work. However, it would be beneficial to mention and discuss more about recent advancements in merging techniques for PEFT.

**Experimental Designs Or Analyses:**

The experimental design is generally strong, incorporating thorough performance comparisons on different sizes of model. The authors evaluate SeedLoRA on multiple datasets, different model sizes (LLaMA2-7B, Mistral-7B, and LLaMA2-13B), and various LoRA configurations.

**Methods And Evaluation Criteria:**

The methods and evaluation criteria align well with the problem at hand. The authors utilize well-established benchmarks (GSM8K, MATH, HumanEval, etc.) and compare SeedLoRA against strong baselines, including vanilla LoRA, full fine-tuning, and alternative model merging methods. The evaluation setup is comprehensive, covering multiple datasets and architectures, and the results are consistently analyzed across different configurations.

**Other Comments Or Suggestions:**

Minor suggestions:
- TIES at line 217 should be cited first.

**Other Strengths And Weaknesses:**

I think the idea and observations presented in the paper are novel, as they address the performance gap by merging LoRA modules trained with different random seeds. The authors provide strong empirical validation with diverse datasets and models, as well as practical relevance for improving PEFT methods. However, a downside of the paper is that the training cost is not mentioned or discussed. Although the authors present strong evidence supporting their proposed method, the theoretical justification could be more rigorous beyond cosine similarity analysis. Additionally, an ablation study on the fusion methods is expected.

**Questions For Authors:**

- How do the merging methods affect the final performance? For example, what is the performance of $\tau_{\text{fused}}$​ alone, $\tau_{\text{fused}} + \tau_{\text{conflict}}$​, and $\tau_{\text{fused}} + \tau_{\text{robust}}​$?
- Would the authors explicitly state the values of σ\sigmaσ used for each dataset and model?
- The algorithm for selecting conflicting dimensions is vaguely described. Could the authors define what “multiple distinct groups” means and explain how to calculate and determine the conflicting dimensions?
- The algorithm for fusing the set of coordinates $\tilde{Z}$ is also unclear. Could the authors explicitly show how $\tilde{Z}$ is calculated??
- Would you compare the training time? What if we fix a seed and train it for n \times longer so that the total training time matches the entire SeedLoRA process?

**Relation To Broader Scientific Literature:**

The paper resides within the existing literature on parameter-efficient fine-tuning (PEFT), model merging, and low-rank adaptation techniques. The discussion effectively differentiates SeedLoRA from related work, such as Model Soup, MoE-based LoRA approaches, and traditional multi-task model merging methods. The paper acknowledges key related works, such as ReLoRA, DoRA, and MultiLoRA, and articulates how SeedLoRA improves upon them.

**Theoretical Claims:**

The paper does not provide theoretical analysis.

---

> ### Author Rebuttal · Authors · 2025-04-01
>
> Thanks for your insightful comments, we carefully address your concerns below.
>
> >W1: However, a downside of the paper is that the training cost is not mentioned or discussed.
>
> While SeedLoRA maintains the same inference memory footprint as vanilla LoRA, we acknowledge that the training computation increases proportionally with the number of seed models. However, since each training process is independent, we can leverage parallelization techniques to accelerate the overall training.
>
> Additionally, we conducted experiments to evaluate SeedLoRA under comparable computational budgets as vanilla LoRA (Line 429). For instance, we compared vanilla LoRA trained for 3 epochs against SeedLoRA with 3 merged models (each trained for only 1 epoch). Our results demonstrate that SeedLoRA achieves superior performance.
>
> >Q1: How do the merging methods affect the final performance?
>
> We try to provide the results of ablation study on LLaMA2-7b fine-tuning on MetaMathQA. We can obtain that these results show both stages are valuable, with their combination yielding optimal performance. Stage 2 alone performs slightly better than Stage 1 alone, and the combined approach consistently improves results across different model architectures.
>
> |Model | stage 1 | stage 2 |  GSM8K  |  MATH  |
> |---------|------|-------|---------|-------|
>  LLaMA2-7b  | $\checkmark$  |    |   67.3  | 16.7
>  LLaMA2-7b  |  | $\checkmark$  |  67.9  | 16.8
>  LLaMA2-7b  | $\checkmark$ | $\checkmark$  |  69.1 | 17.1
>  Mistral-7b  | $\checkmark$  |   |  79.4 | 28.3
>  Mistral-7b  |  | $\checkmark$  |  79.7  |  28.5
>  Mistral-7b  | $\checkmark$ | $\checkmark$  |  80.7   |  28.8
>
> >Q2: Would the authors explicitly state the values of $\sigma$ used for each dataset and model?
>
> For all models and datasets, we determined the threshold $\sigma$ using the top 50\% magnitude of weights across adapters in each layer. This approach provides an adaptive threshold that scales appropriately with the distribution of weight magnitudes in different layers and models.
>
> >Q3: Could the authors define what “multiple distinct groups” means and explain how to calculate and determine the conflicting dimensions?
>
> We determine conflicting dimensions through the following specific process:
>
> For each parameter dimension $j$, we examine the values $\tau_i(j)$ across all n adapters.
> We classify adapters into sign groups based on the sign of their values in dimension $j$.
> A dimension is considered "conflicting" when it meets both of the following criteria:
>
> (1) It contains at least two sign groups (positive and negative).
>
> (2) Each sign group contains at least  $\lfloor n/3 \rfloor$ adapters with magnitude $|\tau_{i}(j)| \geq \sigma$.
>
> This means that for dimension j to be conflicting, there must be a substantial number of adapters (at least one-third of the total) strongly pulling in opposite directions. For example, with 3 adapters, a dimension is conflicting if at least 1 adapter has a large positive value and at least 1 has a large negative value.
>
> >Q4: The algorithm for fusing the set of coordinates $\widetilde{Z}$ is also unclear.
>
> The calculation of the fused coordinate matrix $\widetilde{Z}^{(l)}$ proceeds as follows:
>
>   - After projecting each adapter's moderate parameters $\tau_i^{(l)}$ onto the common subspace to obtain coordinate matrices $Z_i^{(l)}$, we compute the element-wise average of these coordinate matrices: $Z_{\text{avg}}^{(l)} = \frac{1}{n}\sum_{i=1}^{n} Z_i^{(l)}$
>
> - We then examine each element $(m,k)$ in these coordinate matrices across all adapters:
>    \begin{enumerate}
> 	- If the element values $Z_i^{(l)}(m,k)$ have the same sign across all adapters, we retain the average value: $\widetilde{Z}^{(l)}(m,k) = Z_{\text{avg}}^{(l)}(m,k)$
> 	- If there are sign conflicts among adapters, we follow a modified TIES approach:
> $$\widetilde{Z}^{(l)}(m,k) = \frac{1}{|I^+|}\sum_{i \in I^+} Z_i^{(l)}(m,k), \quad \text{if } |I^+| \geq |I^-|$$
> $$\widetilde{Z}^{(l)}(m,k) = \frac{1}{|I^-|}\sum_{i \in I^-} Z_i^{(l)}(m,k), \quad \text{otherwise}$$
>        where $I^+$ and $I^-$ are the sets of adapter indices with positive and negative values for element $(m,k)$, respectively.
>
> This approach preserves agreement between adapters while resolving conflicts by favoring the dominant sign group.
>
> >Q5: Would you compare the training time? What if we fix a seed and train it for n $\times$ longer so that the total training time matches the entire SeedLoRA process?
>
> We have conducted the experiments comparing SeedLoRA with longer training of a single model. In the "Training with More Epochs" section (line 408) and Figure 4(a)-(b), we compare merging 3 LoRA models (each trained for 3 epochs) against a single LoRA model trained for 9 epochs, ensuring the training computation cost is equivalent. Our results show that SeedLoRA outperforms the longer training approach on both mathematical reasoning and code generation tasks.

---

> > ### Comment · Reviewer_7HoA · 2025-04-05
> >
> > I would like to thank the authors for pointing out the lines that address my original questions and for providing detailed responses to my concerns. I maintain my positive view of this paper and will therefore keep my current rating.

---

> > > ### Author Response · Authors · 2025-04-07
> > >
> > > We would like to express our sincere gratitude to Reviewer 7HoA for acknowledging our responses and maintaining your positive evaluation of our work. We appreciate your constructive feedback throughout the review process and will incorporate all suggested changes in the revision of our paper.

---

### Official Review · Reviewer_1ENu · 2025-03-13

**Overall Recommendation:** 3

**Summary:**

The authors propose a method for combining multiple LoRA adapters trained for the same task with different seeds and show it improves performance. The authors’ method consists of (1) identifying and preserving large consistent dimensions or “robust” directions, (2) using a principal-component-like decomposition (SVD / PCA) for ``moderate'' dimensions, and (3) merging these decomposed updates via a thresholded weighted average.

**Claims And Evidence:**

* Single- vs. Multi-Task Model Differences: They assert that they have examined differences between single-task and multi-task model merging (Section 3.3). However, the presented evidence mostly focuses on single-task settings (and the multi-task data come only from a brief cosine similarity comparison). This makes the claim of “understanding differences” (their stated contribution (i)) less thoroughly supported. The study would benefit from more extensive experiments or references to truly confirm those differences.

* Performance Gains: Their experiments do convincingly demonstrate that their adapter merging method yields improvements. This is generally well-supported by results on math and coding tasks.

**Essential References Not Discussed:**

This is of course related to MoE and model merging, which I am not that familiar with. Possibly more thorough discussion of SWA-like arguments (which they partially reference) could reinforce the “wider optimum” viewpoint in merging. But otherwise, no major omissions stand out.

**Experimental Designs Or Analyses:**

The experimental design is straightforward, benchmarks are well-known, and baselines reasonable. The results seem reproducible. Since across seed variation is one of the fundamental claims of the papers, it would be helpful if the authors added e.g. in table 2 and 3 how when reporting lora if that is the average across seeds and how meny where used (and perhaps stds). Also, in Figure 2, it is unclear whether the “cosine similarity” reported is computed for a specific layer or is an average across all layers/adapters—further clarity in the text would help.

**Methods And Evaluation Criteria:**

The experimental settings and metrics used (GSM8K, MATH, HumanEval, MMLU, etc.) are standard, and the results are generally comprehensive.

**Other Comments Or Suggestions:**

Cite TIES in 3.3. Make figure and table captions more comprehensive.

**Other Strengths And Weaknesses:**

Although their proposed method is more general and is not restricted LoRA adapters, they demonstrate its  utility in this setting which is relevant in practice.

**Questions For Authors:**

N/A

**Relation To Broader Scientific Literature:**

The idea of ensembling models across runs to leverage the sources of randomness during training is not new, but there are few works that combine it with PEFT of LLMs. Most existing works (Multi LoRA, IterIS) focus on the multi-task, while this paper focuses on single tasks.

**Theoretical Claims:**

There is no deep theoretical result presented in the paper. Most claims about the “differences” between single-task and multi-task models, or about why merging addresses complementary subdomains, remain at an intuitive or empirical level. This is okay since this is as an applied paper, though ensuring clarity of definitions (e.g., the rank constraints, the threshold) would strengthen the argument. The paper could also benefit from a more clear notation (e.g. tau denotes both the threshold and the weight updates)

---

> ### Author Rebuttal · Authors · 2025-04-01
>
> Thanks for your constructive and inspiring feedback, we carefully address your concerns below.
>
> >Clarification on seed reporting and cosine similarity calculations
>
> Thank you for your feedback on the experimental design and reproducibility. We agree that additional clarity on seed variation would strengthen our presentation.
>
> (1) Reporting of LoRA results in Tables 2 and 3
>
> In Tables 2 and 3, we report individual results for each seed (seed=11, seed=42, seed=202) rather than averages. Each column represents the performance of a single LoRA model trained with that specific seed. We chose to present the individual seed results rather than averages to highlight the performance variation across seeds, which is a key motivation for our SeedLoRA approach.
>
> (2) Clarification on cosine similarity in Figure 2
>
> Regarding Figure 2, the cosine similarity values shown are computed as averages across all layers of the models. Specifically, for each adapter pair, we first computed the cosine similarity between corresponding parameters in each layer, then averaged these similarities across all layers to obtain the single value shown in the figure.
>
> >Possibly more thorough discussion of SWA-like arguments (which they partially reference) could reinforce the “wider optimum” viewpoint in merging.
>
> Thank you for your insightful comment regarding the potential connection between SeedLoRA and SWA-style techniques.
>
> While both SWA and SeedLoRA involve combining model weights, their underlying assumptions and practical implications differ substantially. SWA averages model weights sampled along the same optimization trajectory, typically assuming that these checkpoints lie within the same basin of attraction in the loss landscape. This allows SWA to converge to flatter optima, often associated with better generalization and robustness.
>
> In contrast, SeedLoRA merges multiple LoRA adapters trained from different random seeds, which often leads to models residing in different regions of the parameter space—potentially even in distinct basins. As a result, direct weight averaging, as in SWA, can be ineffective or even detrimental in this context.

---

### Official Review · Reviewer_CWvH · 2025-03-14

**Overall Recommendation:** 3

**Summary:**

The paper introduces SeedLoRA, an approach to improving LoRA fine-tuning for LLMs. SeedLoRA is based on the observation that multiple LoRA models trained on the same task with different random seeds can have complementary performance. It uses a two-stage approach to merge different LoRA adapters, first identifying conflicting/robust parameters, and then performing subspace fusion via SVD to merge the remaining parameters. Experiments on Llama-2-7B/13B and Mistral-7B demonstrate improvements in math reasoning and code generation, achieving near full fine-tuning performance.

**Claims And Evidence:**

Yes

**Essential References Not Discussed:**

NA

**Experimental Designs Or Analyses:**

Yes

**Methods And Evaluation Criteria:**

Yes

**Other Comments Or Suggestions:**

NA

**Other Strengths And Weaknesses:**

Strengths
1. SeedLoRA achieves comparable performance to full fine-tuning and shows significant improvement in performance over vanilla LoRA over a wide range of tasks.
2. The idea of handling robust/contradicting parameters first and then merging is interesting and works well in the examples.


Weaknesses
1. The experiments mainly focus on merging models with rank 8, with only a single comparison against a rank 24 model. This raises questions about the method's effectiveness across different ranks. Given that higher-rank (rank 32/64) LoRA adapters are commonly used (e.g., various models on Huggingface), it remains unclear whether SeedLoRA's improvements are consistent across higher-rank LoRAs or are limited to low-rank settings.
2. The ablation experiment comparing merging adapters to longer training (9 epochs) lacks granularity. Training high-rank models for fewer epochs might offer better compute-performance tradeoffs than training a low-rank model for 9 epochs, especially if overfitting occurs with extra training.
3. The paper does not quantify the proportion of parameters classified as robust/conflicting during stage 1, nor does it justify the threshold $\sigma $. What percentage of parameters fall into robust/conflicting categories, and how does this vary across tasks/layers? How is $\sigma$ chosen here, is it task/layer dependent, and could adaptive thresholds improve results? If the proportion of robust/conflicting parameters is very small, it would be insightful to include an ablation on whether stage 1 is needed or not. For example, does merging all parameters only through subspace fusion degrade performance?
4. The experiments are limited to merging 3 seed models. Based on the assumption that different adapters have different comparative strengths, it would be interesting to see if performance continues to improve with more seed models. A follow-up question is when does the performance stop to scale or even drop with more seeds added, or is 3 seeds the maximum number of seeds that works with this method?

**Questions For Authors:**

See strengths and weaknesses.

**Relation To Broader Scientific Literature:**

The key contributions of SeedLoRA build on several lines of research in PEFT and model merging. SeedLoRA directly extends the LoRA framework, which freezes pre-trained weights and injects trainable low-rank matrices. Some more recent works, like ReLoRA/DoRA/MoRA, improve LoRA's performance and can be plugged directly into SeedLoRA as this work focuses on post-training fusion rather than specific architectural changes. SeedLoRA also takes inspiration from Model Soup and methods like TIES/DARE/SWA and developed a two-stage adapter merging approach while showing superior performance over these methods.

**Theoretical Claims:**

NA

---

> ### Author Rebuttal · Authors · 2025-04-01
>
> Thanks for your constructive and inspiring feedback, we carefully address your concerns below.
>
>  >W1: Effectiveness across different ranks.}
>
> To directly address this concern about the scalability of SeedLoRA across different rank settings, we have conducted comprehensive additional experiments with higher-rank LoRA adapters (r=16, 32, 64 and 96) on the MetaMathQA benchmark.
>
> - When merging 3 models (LLaMA2-7b):
>
> |  | GSM8K |  |   |  | MATH |  |  |  |
> |---------|------|-------|---------|-------|---------|------|---------|------|
> |Rank |seed=11 |seed=42 | seed=202 | SeedLoRA | seed=11 | seed=42 | seed=202 | SeedLoRA |
> |LoRA (R=8)| 64.0 | 63.8 | 64.1 | **68.4** | 15.3 | 15.3 | 14.9 | **17.3**
> |LoRA (R=32)| 65.4 | 66.0 | 66.9 | **70.3** | 16.0 | 16.7 | 16.4 | **17.8**
> |LoRA (R=64)| 65.7 | 65.5 | 66.6 | **69.2** | 16.2 | 16.5 | 16.5 | **17.4**
> |LoRA (R=96)| 66.7 | 66.0 | 66.4 | **70.0** | 16.7 | 17.2 | 16.9 | **17.1**
>
> These results clearly demonstrate that SeedLoRA consistently improves performance across all tested rank settings. This confirms that SeedLoRA's effectiveness is not limited to low-rank settings but extends to the higher-rank adapters.
>
> >W2: Merging adapters to longer training (9 epochs) lacks granularity.
>
> To address this concern, we have conducted additional experiments comparing SeedLoRA with equivalent or higher-rank single LoRA adapters:
>
> |Model |Task |LoRA (R=24) | SeedLoRA (3*(R=8)) | LoRA (R=96) | SeedLoRA (3*(R=32)) |
> |---------|------|-------|---------|-------|---------|
> |LLaMA2-7b| GSM8K | 64.9 | **68.4**  | 66.7  | **70.0**
> |LLaMA2-7b| MATH | 16.3  |  **17.3** |  16.7 |  **17.1**
>
> These results demonstrate that SeedLoRA offers superior performance compared to single higher-rank alternatives. SeedLoRA (merging three r=8 adapters) achieves +3.5\% better performance on GSM8K compared to a single r=24 LoRA adapter.  Moreover, the improvement scales to very high ranks, with SeedLoRA (merging three r=32 adapters) outperforming a single r=96 LoRA adapter by +3.3\% on GSM8K.
>
> >W3: The proportion of parameters classified as robust/conflicting during stage 1, nor does it justify the threshold .
>
> To address these concerns, we've analyzed the proportion of parameters classified in each stage across different layers.
>
> (1) Layer-wise parameter distribution analysis
>
> We measured the proportion of parameters classified across different layers and tasks:
>
> - LLaMA2-7b on MetaMathQA:
>
> |Layer |robust (stage 1) | conflicting (stage 1) | stage 2 |
> |---------|------|-------|---------|
>  Attention Q  | 9.0 \%  | 6.4 \%  |  84.6 \%
>  Attention K   | 9.3 \%  | 6.7 \%  |  84.0 \%
>  Attention V  | 5.2 \%  | 8.5 \%  |  85.6 \%
>  MLP up\_proj  | 6.9 \%  | 7.5 \%  |  86.0 \%
>  MLP down\_proj  | 6.2 \%  | 7.8 \%  |  86.3 \%
>
> Overall, approximately 13-16\% of parameters are classified as robust or conflicting in Stage 1, with variations across layer types.
>
> (2) Threshold selection and ablation study
>
> The threshold was determined using the top 50\% magnitude of weights across adapters in each layer. This approach provides an adaptive threshold that scales appropriately with the distribution of weight magnitudes in different layers and models.
>
> To verify the importance of Stage 1, we conducted an ablation experiment:
>
> - LLaMA2-7b and Mistral-7b on MetaMathQA
>
> |Model | stage 1 | stage 2 |  GSM8K  |  MATH  |
> |---------|------|-------|---------|-------|
>  LLaMA2-7b  | $\checkmark$  |    |   67.3  | 16.7
>  LLaMA2-7b  |  | $\checkmark$  |  67.9  | 16.8
>  LLaMA2-7b  | $\checkmark$ | $\checkmark$  |  69.1 | 17.1
>  Mistral-7b  | $\checkmark$  |   |  79.4 | 28.3
>  Mistral-7b  |  | $\checkmark$  |  79.7  |  28.5
>  Mistral-7b  | $\checkmark$ | $\checkmark$  |  80.7   |  28.8
>
> These results show both stages are valuable, with their combination yielding optimal performance.
>
> - Merging all parameters through subspace fusion: we conducted additional experiments about SeedLoRA (rank=8) with only subspace fusion for all parameters, which resulted in 67.5\% on GSM8K and 16.4\% on MATH (SeedLoRA: 69.1\% on GSM8K and 17.1\% on MATH).
>
> >W4: If performance continues to improve with more seed models.
>
> To investigate scaling behavior when merging additional models, we conducted experiments merging up to 6 different seed models:
>
> |Task | Rank | SeedLoRA (2)  | SeedLoRA (3)  | SeedLoRA (4)  | SeedLoRA (5)  | SeedLoRA (6)  |
> |---------|------|-------|---------|-------|---------|-------|
>  GSM8K  | LoRA (R=8)  | 67.7 | 68.4 | 69.7 | 69.6 | 69.8
>  MATH  | LoRA (r=8) | 16.6  | 17.3 | 17.1 | 16.6 | 17.1
>
> These results show significant initial gains when merging 2 models (+3.7\% on GSM8K), with continued improvements up to 4 models (+5.7\% on GSM8K). Beyond 4 models, we observe diminishing returns without performance degradation.

---

> > ### Comment · Reviewer_CWvH · 2025-04-09
> >
> > Thank you for your detailed response, some of my concerns have been properly addressed. I will update my recommendation.

---

> > > ### Author Response · Authors · 2025-04-09
> > >
> > > We sincerely appreciate Reviewer CWvH for the thoughtful evaluation and valuable feedback. We will incorporate all suggested changes in our revision. Thank you for your time and expertise in reviewing our submission.

---

### Decision · Program_Chairs · 2025-05-01

**Decision:**

Accept (poster)

**Comment:**

This paper introduces SeedLoRA, a novel post-training method for improving LoRA performance by merging adapters trained with different random seeds. The core insight is that different initializations lead LoRA to capture complementary aspects of a task, and their combination yields a stronger model. All reviewers found the idea interesting and the empirical results, showing significant gains over vanilla LoRA and often matching full fine-tuning on challenging benchmarks, compelling.

Initial reviews were unanimously Weak Accept, raising concerns about scalability (rank, number of seeds), algorithmic clarity/justification, comparison fairness/cost, and theoretical grounding. The authors provided a thorough rebuttal with substantial new experiments and clarifications that addressed the majority of these points effectively. Reviewers CWvH and 7HoA explicitly acknowledged that their concerns were addressed, with CWvH indicating an intent to raise their score. While theoretical backing remains light, the strong empirical validation and successful rebuttal addressing key experimental and methodological questions support acceptance.